# Anion-Binding Properties of Short Linear Homopeptides

**DOI:** 10.3390/ijms25105235

**Published:** 2024-05-11

**Authors:** Matija Modrušan, Lucija Glazer, Lucija Otmačić, Ivo Crnolatac, Nikola Cindro, Nikolina Vidović, Ivo Piantanida, Giovanna Speranza, Gordan Horvat, Vladislav Tomišić

**Affiliations:** 1Department of Chemistry, Faculty of Science, Horvatovac 102a, 10000 Zagreb, Croatia; mmodrusan@chem.pmf.hr (M.M.); lucija.glazer@chem.pmf.hr (L.G.); lotmacic@chem.pmf.hr (L.O.); ncindro@chem.pmf.hr (N.C.); vtomisic@chem.pmf.hr (V.T.); 2Department of Organic Chemistry and Biochemistry, Ruđer Bošković Institute, Bijenička Cesta 54, 10000 Zagreb, Croatia; icrnolat@irb.hr (I.C.); ivo.piantanida@irb.hr (I.P.); 3Faculty of Biotechnology and Drug Development, University of Rijeka, Radmile Matejčić 2, 51000 Rijeka, Croatia; nikolina.vidovic@biotech.uniri.hr; 4Department of Chemistry, University of Milan, Via Golgi 19, 20122 Milan, Italy; giovanna.speranza@unimi.it

**Keywords:** linear peptides, anions, complexation, stability constant, structure, fluorimetry, circular dichroism, ITC, ^1^H NNR, molecular dynamics

## Abstract

A comprehensive thermodynamic and structural study of the complexation affinities of tetra (**L1**), penta (**L2**), and hexaphenylalanine (**L3**) linear peptides towards several inorganic anions in acetonitrile (MeCN) and *N*,*N*-dimethylformamide (DMF) was carried out. The influence of the chain length on the complexation thermodynamics and structural changes upon anion binding are particularly addressed here. The complexation processes were characterized by means of spectrofluorimetric, ^1^H NMR, microcalorimetric, and circular dichroism spectroscopy titrations. The results indicate that all three peptides formed complexes of 1:1 stoichiometry with chloride, bromide, hydrogen sulfate, dihydrogen phosphate (DHP), and nitrate anions in acetonitrile and DMF. In the case of hydrogen sulfate and DHP, anion complexes of higher stoichiometries were observed as well, namely those with 1:2 and 2:1 (peptide:anion) complexes. Anion-induced peptide backbone structural changes were studied by molecular dynamic simulations. The anions interacted with backbone amide protons and one of the *N*-terminal amine protons through hydrogen bonding. Due to the anion binding, the main chain of the studied peptides changed its conformation from elongated to quasi-cyclic in all 1:1 complexes. The accomplishment of such a conformation is especially important for cyclopeptide synthesis in the head-to-tail macrocyclization step, since it is most suitable for ring closure. In addition, the studied peptides can act as versatile ionophores, facilitating transmembrane anion transport.

## 1. Introduction

The scientific interest in the development of selective ion receptors has undergone a continuous rise during the last few decades, particularly in the design of anion receptors [1,2]. Among the various supramolecular anion receptors, peptides are one of the most interesting compounds due to their natural abundance, variability of subunits, and their relatively straightforward synthesis [3,4,5]. Furthermore, research on the physicochemical characteristics of peptide–anion complexes offers an insight into the interactions between substrates and amino acids comprising protein active sites and is useful in the study of enzymatic reaction mechanisms. Peptide backbone amide groups can form several ion binding motifs such as *nest* [6] for anion and *catgrip* [7] or *niche* [8] for cation binding, which are often found in important parts of proteins (active sites and anion-binding proteins). In the large pool of peptide ionophore candidates [9,10,11], cyclic peptides were found to have the best ion-binding properties [12,13], i.e., these compounds exhibit enhanced binding affinity and selectivity towards substrates when compared to their more flexible linear analogs. Moreover, their metabolic stability and high bioavailability make them perfect model compounds for the study of antibiotic, anticancer, and membrane transport properties [14,15].

The synthesis of small cyclopeptides often ends up with low reaction yields due to the unfavorable quasi-cyclic conformational reorganization of linear peptides, that is the ability of a linear precursor to bring its reactive termini in the close spatial proximity prior to ring closure. The cyclization step is accompanied by a very high level of oligomerization and epimerization, ending with macrocyclization yields as low as a few percent. Over the years, many strategies have been developed to avoid undesired reactions and to direct macrocyclization, but most of them are highly dependent on the peptide’s secondary structure [16]. An elegant way to carry out ring closure, independently on the peptide’s structure, is to perform the reaction with the help of templating agents that can bind to linear precursors. The most commonly used templates are simple inorganic ions, particularly metal cations, due to their spherical size and ability to be coordinatively bound to peptide functional groups. More recently, anion templating properties for the cyclization of anion receptors have been investigated as well [17,18,19,20]. This was especially important in the case of the cyclization of smaller peptides comprising four to six amino acids, where it was believed that alkali metal cations promote cyclization [21,22]. However, our previous work demonstrated that chloride ions act as templating agents in this synthetic step by the formation of hydrogen bonds with the amide protons of the peptide backbone, thus forcing the peptide into a pseudocyclic structure [23]. Such a finding, together with the scarcity of literature data on the binding of anions by linear peptides [14,24], encouraged us to investigate these interesting peptide properties in the hope of bringing us one step closer to the pursuit of finding the optimal templating agents for small-peptide macrocyclization.

Furthermore, the number of hydrogen bonds formed in the peptide–anion complex is directly related to the length of the peptide chain, a structural parameter that can also play an important role in the strength and selectivity of peptide–anion binding. The existence of such an effect would be interesting in the context of cyclopeptide synthesis.

In this work, we investigated the chain-length effect on the anion binding to the short linear homopeptides comprising four to six phenylalanine subunits. For this purpose, we synthesized methyl esters of tetra (**L1**), penta (**L2**), and hexaphenylalanine (**L3**) (Figure 1) and thermodynamically characterized their complexation with a number of anions in acetonitrile (MeCN) and *N*,*N*–dimethylformamide (DMF) by means of several analytical methods. In the cyclization step of the cyclopeptide synthesis, linear peptide precursors are unprotected at the N-terminus and have a coupling reagent attached to the C-terminus. The linear peptide derivatives used in this work represent those used in cyclopeptide synthesis, as they are unprotected at the N-terminus and have a carboxylate derivative group at the C-terminus. The additional reason for the selection of these particular peptides lies in their fluorescence ability. Acetonitrile was chosen due to its poor anion solvation properties that promote the formation of stable peptide–anion complexes, whereas DMF was used due to the fact that the macrocyclization synthetic step is carried out in this solvent. The structure of peptide–anion complexes was studied computationally by molecular dynamic (MD) simulations with explicit solvent molecules.

## 2. Results and Disscusion

**Synthesis of compounds L1, L2, and L3.** The compounds **L1** and **L2** were prepared from commercially available compounds according to the synthetic pathway described elsewhere [23]. Hexapeptide **1** was prepared by the peptide coupling of compound **L2** with CbZ-protected phenylalanine (Figure 2). The transfer hydrogenation of **1** gave the desired compound **L3** (Appendix A). Transfer hydrogenation was used in the last reaction instead of the hydrogenation with H_2_ at RT to affect the solubility of compound **1** (which is low at room temperature) by carrying out the reaction at a higher temperature and generating H_2_ in situ.

**Anion complexation by peptides in MeCN.** The formation of **L1**, **L2**, and **L3** anion complexes in acetonitrile was studied by spectrofluorimetry, microcalorimetry, and ^1^H NMR spectroscopy. From these measurements, we determined not only the stability constants of peptide–anion complexes but also the complexation enthalpies and entropies (Table 1 and Table 2). The titration curves corresponding to chloride, bromide, and nitrate anions were successfully processed assuming a 1:1 stoichiometry of the formed complexes. In the case of HSO_4_^−^ and H_2_PO_4_^−^, a complexation giving 1:2 (peptide:anion) complexes was included in the model as well. An example of the spectrofluorimetric titration of **L1** with tetrabutylammonium chloride is presented in Figure 1, whereas the titrations with other anions are shown in the Appendix A. The fluorescence of the benzyl group of phenylalanine was most intense for the excitation at 260 nm. For all investigated peptides, the addition of chloride, hydrogen sulfate, or dihydrogen phosphate anions led to an increase in the fluorescence intensity (Figure 1, Appendix A) as opposed to bromide and nitrate anions, where fluorescence quenching was observed (Appendix A).

The results of the ^1^H NMR measurements regarding the anion–peptide complex formation supported those obtained by spectrofluorimetry, with the stability constants determined by both methods being, in most cases, in satisfactory agreement (Table 1). The most significant changes in the NMR spectra of all three receptors upon anion binding were observed for the signals of the backbone amide, C-α, and methyl ester protons, most probably due to their direct participation in hydrogen bonding with complexed anions and peptide chain reorganizations (Figure 2, Appendix A).

Additionally, we performed CD titrations to follow the structural changes of amide groups that occur by anion complexation and to test this method for reliability for the determination of stability constants. The titration results are shown in Figure 3, Appendix A. The most significant differences in ellipticity in these titrations were observed at 237 nm at the π* *← n* transition band of the amide group. Since the peptide CD signal is directly influenced by the mutual orientation of these groups, the change in spectra indicates the conformational transition upon binding. In the case of 1:1 peptide–anion complexes, molar CD spectra have a positive signal at 237 nm as opposed to the strongly negative signal of free peptides (Figure 4, Appendix A). Interestingly, the shape of the free receptor spectra and spectral changes observed in CD titrations of linear peptides are similar to the ones observed in the chloride binding to a linear heptapeptide in chloroform [25] and are exactly the opposite of the changes we observed in the CD titrations of cyclopentaphenylalanine with the same anions, where the free receptor spectrum was flat at the 240 nm region, and the anion binding induced the appearance of the negative peak [26]. Stability constants of **L**-Cl^−^ and **L**-Br^−^ complexes obtained by CD titrations are given in Table 1 and are in satisfactory agreement with the ones determined by other methods. To the best of our knowledge, this is the first account of the stability constant determination of linear peptide–anion complexes by circular dichroism. In CD titrations with Br^−^, a partial overlap of the anion and peptide signals was observed, which reduced the reliability of the equilibrium constant determination. We were unable to obtain reliable stability constant estimates for the hydrogen sulfate and DHP peptide complexes from the CD titrations (Appendix A), although the observed spectral changes indicated that anion binding occurred.

The analysis of the stability constants of peptide–anion complexes gives way to the thermodynamic and structural description of the complexation process. The affinities of all three peptides towards chloride anions are similar, although peptides differ in chain length. This suggests that the chloride anion size is equally thermodynamically compatible with the binding sites of tetra-, penta-, and hexapeptides. To better understand the effects that drive the binding of chloride anions, microcalorimetric titrations were conducted. By these titrations, we were able to determine standard reaction enthalpic and entropic contributions to the standard reaction Gibbs energy (Table 2 and Figure 5, Appendix A). The ITC measurement results show an enthalpically favorable formation of the **L**-Cl^−^ complexes for all three peptides. Interestingly, the reaction enthalpy for the **L1** and **L2** chloride complexation is similar, although **L2** has an additional backbone amide group at the binding site. The entropic part is more favorable for the formation of the **L2**Cl^−^ complex, which is also a bit counterintuitive regarding the chain length. In the case of the larger hexapeptide **L3**, the enthalpic contribution is almost twice as exothermic than for the peptides **L1** and **L2**, which could be at least partially contributed to the highest number of backbone amide groups coordinating the anion. The reaction entropy of the **L3**Cl^−^ formation is significantly unfavorable, which is in line with the chain reorganization and higher loss of the degrees of freedom for larger linear molecules. These findings are complemented by the structural characterization of free peptides and their complexes performed by MD simulations (Table 3 and Figure 5b, Appendix A). In the simulations of the free peptides, the formation of intramolecular hydrogen bonds was observed. In the MD simulation of the free **L1** peptide in 0.9 of acetonitrile, intramolecular hydrogen bonds were observed on average, as well as for 1.5 for **L2** and 2.3 in the case of **L3**. These interactions surely impact the thermodynamics of anion complexation in two ways: enthalpically, due to a hydrogen bond disruption upon complexation, and entropically, because of the free peptide chain structuring. However, these effects cannot be directly correlated with the ITC data. The simulations of the peptide–chloride anion complexes indicate that the anion is, on average, coordinated by nearly all backbone amide protons of **L1**–**L3** and with one of the amine protons as well (Table 3). The representative structures of the **L**-Cl^−^ complexes clearly show that peptides adopt a quasi-cyclic structure upon anion coordination, which is suitable for the macrocyclization step in cyclopeptide synthesis. This can be correlated with the experimental results of CD titrations, where the molar spectra of all three **L**-Cl^−^ complexes are alike at the 240 nm region (Figure 4, Appendix A). This indicates that the backbone of all three peptides is reorganized in a similar manner, as observed by the MD simulations.

Bromide anion forms 1:1 complexes with all three peptides. There are no significant differences in the stability of these complexes regarding the chain length, and the binding of the bromide anion is slightly weaker relative to the complexation of chloride. This is probably due to the lower charge density of bromide ions, which results in a weaker hydrogen bonding between peptides and anions. No ITC measurements for these processes were performed due to the low stability of the bromide–peptide complexes, which would require a high salt concentration that would lead to high dilution heats and/or the influence of ionic strength on the binding process.

The data obtained by the ITC titrations indicate that in addition to the **L**–HSO_4_^−^ complexes, the 1:2 **L**–(HSO_4_)_2_^2−^ were formed as well (Table 2 and Appendix A). Hydrogen sulfate does not form dimers in a solution, but upon binding to the supramolecular host, it can form dimers with the already bound one [27,28,29]. Spectrofluorimetric titrations could be processed by the model used for the ITC data fitting. By assuming this model, the fitting process gave excellent agreement between the observed and calculated data, although the molar spectra of free peptides and their 1:1 complexes were similar (Appendix A), which reduced the reliability of the results obtained by this method. The results of these titrations indicate that all three phenylalanine peptides form very stable complexes with hydrogen sulfate, whereby the stability constant is about two orders of magnitude higher than for the corresponding chloride and bromide anion complexes. The stability of the **L**–HSO_4_^−^ and **L**–(HSO_4_)_2_^2−^ complexes is similar for all peptides, although the enthalpic and entropic contributions are relatively different. The formation of both types of complexes between peptides and hydrogen sulfate is enthalpy driven. The reaction yielding a 1:1 complex is the most exothermic in the case of pentapeptide, whereas formation of a 1:2 complex is the most exothermic in the case of hexapeptide. The entropic part of the reaction Gibbs energy for the 1:1 complex formation varies between peptides, which is probably due to the interplay between the loss of degrees of freedom during chain reorganization and the degree of anion desolvation upon complex formation. The binding of the second hydrogen sulfate mostly proceeds with a slightly negative cooperativity because of the highly unfavorable entropic contribution due to the loss of anion translational degrees of freedom and the possible reorganization of peptide chains as indicated by the changes in the CD spectra (Appendix A).

The complexation of dihydrogen phosphate (DHP) is equivalent to the binding of hydrogen sulfate in terms of stoichiometry, except for the fact that a DHP anion forms dimers in an acetonitrile solution [30,31]. The 1:1 type of complex of all three peptides has a similar stability as the **L**–HSO_4_^−^ complexes, whereas the binding of the second DHP is more pronounced than that of hydrogen sulfate but still with a negative cooperativity. A moderate preference for DHP for the longer peptide chains is seen in the increase in the stability constants of both the 1:1 and 1:2 DHP complexes with the number of peptide subunits. The obtained thermodynamic data (Figure 6, Appendix A) cannot be directly correlated with the chain length. The 1:1 complex formation is most exothermic for the **L2**–DHP^−^ complex, while the reaction entropy contributes most to the stability in the case of **L3**–DHP^−^.

It is also interesting to compare the anion receptor properties of the linear pentaphenylalanine peptide **L2** and its cyclic derivative cyclopentaphenyalanine, which we studied in our previous research [26]. The stability constants of chloride and bromide complexes with cyclic peptide in acetonitrile are about 3.5 orders of magnitude higher than for the corresponding complexes with **L2**. This suggests that the cyclic pentapeptides are way better receptors of monoatomic anions than their linear analogues. Surprisingly, the stability constants of hydrogen sulfate and DHP complexes are nearly similar for both peptide forms. The origin of this effect needs further investigation, but it most probably lies in the presence of multiple hydrogen bond acceptor sites of oxo anions that could bind similarly to the backbone amide groups of linear and cyclic pentaptides.

The representative structures of the free peptides and their anion complexes obtained by MD simulations are shown in Figure 7, Appendix A. The results of the structural analyses performed on the trajectories of the complexes are given in Table 3. The MD results show a linear conformation of the free peptide, whereas the binding of the anion forces the change of peptide conformation into a cyclic structure where the two termini are closer, which can facilitate the macrocyclization reaction. The complex formation occurs through the formation of hydrogen bonds between the backbone amide and amine hydrogens and anions (oxygen atoms in the case of oxo anions). In all complexes, one amine hydrogen bond is formed, whereas the number of amide hydrogen bonds depends on the peptide length. There is no significant difference in the total number of hydrogen bonds in the series of anion complexes of a particular peptide, which could suggest that desolvation and its extent plays an important role in determining the stability of complexes. Additionally, this means that the linear peptide backbone is flexible enough and can easily adjust to the various shapes and sizes of anions. The analysis of the *φ* and *ψ* torsion angles of the peptide complexes in acetonitrile (Appendix A) indicates that the peptide backbone changes upon complexation, and in the case of **L1** and **L2**, complexes with chloride and bromide anions have the sharpest peaks in angle distributions, suggesting a rigid structure of these complexes. Also, the distance between amine nitrogen and ester carbon atoms (end-to-end distance, Appendix A) is reduced by 30–50% in the complexes compared to the free peptides, which is in line with the anion-templating effect in the linear peptide cyclization step. The peptide–anion interaction energies are favorable for all complexes (Appendix A), but there is no clear correlation to the experimentally determined complex stabilities. There is an obvious decrease in the peptide–solvent interactions after complexation, which can be attributed to the desolvation of the binding site, and this effect is positively related to the chain length (Appendix A). Complexed anions have non-negligible interactions with solvent molecules, indicating that their binding sites are partially desolvated. We were unable to obtain stable structures of the anion dimer complexes due to limitations of the force field parameters or the classical level of theory used for the intermolecular interactions.

**Anion complexation by peptides in DMF.** Anion binding to short linear peptides in DMF is highly significant, since that is the solvent in which the ring closure reaction in cyclopeptide synthesis is performed. Due to the overlap of DMF absorption and the phenylalanine excitation spectrum, spectrofluorimetry had to be excluded as a method for stability constant determination. The stability of peptide–anion complexes comprising halide and nitrate anions are a bit lower in DMF than in acetonitrile (Table 4, Appendix A). The transfer Gibbs energy of the free chloride and bromide anions from acetonitrile to DMF is 5–6 kJ mol^−1^ [32], which should increase the stability constants of the peptide–anion complexes in DMF by about an order of magnitude compared to MeCN. This means that the solvation and conformational changes in free peptides and their anion complexes govern the thermodynamics of complexation. The structural characteristics of chloride and bromide complexes in DMF obtained by MD are similar to the corresponding species in acetonitrile (Table 5 and Appendix A). The conformations of free peptides in these solvents are similar with respect to intramolecular hydrogen bonding (average numbers in DMF are **L1**, 0.9; **L2**, 2.5; and **L3**, 2.3), indicating that the difference in the solvation of free and/or complexed peptides in acetonitrile and DMF is a dominant contribution to the observed difference in complex stabilities. The complexation of chloride in DMF is entropically driven with an unfavorable enthalpic contribution to the reaction Gibbs energy. This is in contrast with the complexation of this anion in acetonitrile, where this process is exothermic and mostly enthalpically controlled. The transfer enthalpy of chloride from acetonitrile to DMF is only about −2 kJ mol^−1^ [32], which cannot account for the overall differences in reaction enthalpies. This again suggests that the solvation energetics of free and complexed peptides is different in these solvents.

In the titrations with hydrogen sulfate in DMF, only a 1:1 type of complex was observed for all peptides by ^1^H NMR and microcalorimetry (Table 4 and Appendix A), whereby the results of both methods agree well. The stability constants of the **L**-HSO_4_^−^ complexes are around two orders of magnitude lower than in acetonitrile, although the complexation is more exothermic in DMF. This decrease in stability is a consequence of a very unfavorable entropic contribution. Hydrogen sulfate anion prefers to bind to longer peptide chains, and the stability constant of the **L3**HSO_4_^−^ complex is an order of magnitude larger than that of **L1**HSO_4_^−^.

The complexation of the DHP anion with **L1**, **L2**, and **L3** results in three types of complexes: 1:1 with stoichiometry, 1:2 with a DHP dimer, and 2:1 when the DHP anion is sandwiched between the two peptide chains (Table 4 and Appendix A). The thermodynamic stability of the DHP anion complexes increases with the chain length for all types of complexes (Table 4). This is mostly due to differences in the complexation enthalpies, although in some cases the reaction entropies play a part in the overall stability of the **L**–DHP complexes.

The stability constants of the **L**–NO_3_^−^ complexes in DMF were determined by ^1^H NMR titrations (Table 4 and Appendix A). Studied peptides **L1**–**L3** weakly bind nitrate anions in DMF, and the stability of corresponding complexes is lower than that with other studied anions.

Molecular dynamic simulations of the **L1**, **L2**, and **L3** anion complexes in DMF were also carried out. It was found that all 1:1 complexes adopt quasi-cyclic structures, where the anion was coordinated by almost all amide protons and one of the amine protons, as was the case in acetonitrile (Table 5 and Appendix A). The structure of the sandwich-type 2:1 DHP–peptide complexes was also investigated by simulations. The DHP anion was found to be indeed situated in between two peptide chains in quasi-cyclic conformations, giving rise to an anion–peptide sandwich assembly (Appendix A). The additional analysis of the MD data in DMF includes the *φ* and *ψ* torsion angle distributions (Appendix A), the distances between amine nitrogen and ester carbon atoms (end-to-end distance, Appendix A), and peptide–anion interaction energies (Appendix A). Similar conclusions can be drawn for the structure and energetics of the peptide–anion complexes in DMF as in acetonitrile, although the dihedral angle distributions of **L2** and **L3** chloride and bromide complexes are a bit less sharp in DMF. This is probably due to the hydrogen bonding properties of the solvent molecules that can interact with peptide backbone amide groups and disrupt the chain conformation in anion complexes.

## 3. Materials and Methods

### 3.1. Synthesis

All amino acid precursors and promoting reagents used for the synthesis of peptides **L1**, **L2**, and **L3** were obtained from Carbolution Chemicals GmbH, St. Ingbert, Germany and were used without further purification. The solvents used for synthetic procedures, namely DMF (Fisher chemical, Pittsburgh, PA, USA, 99.5%, Analytical gradient grade) and MeOH (J.T. Baker, Phillipsburg, NJ, USA, (Ultra) Gradient HPLC Grade), were also used without further purification. NMR spectra of all synthetic intermediates and final products were recorded by means of a Bruker Ascend 400 spectrometer (Billerica, MA, USA) with TMS as an internal standard. The high-resolution mass spectrometry (HRMS) spectra of synthetic products were recorded on a Thermo Fisher Q Exactive ESI orbitrap mass spectrometer. Compounds **L1** and **L2** were prepared according to the procedure published by Vidović et al. [23], whereas the synthesis of **L3** as a new compound starting from **L2** will be described in detail.

### 3.2. Synthesis of N-(carbobenzyloxy)-l-phenylalanyl-l-phenylalanyl-l-phenylalanyl-l-phenylalanyl-l-phenylalanyl-l-phenylalanine Methyl Ester (1)

*N*-(carbobenzyloxy)-l-phenylalanine (485 mg, 1.62 mmol) and HOBt (300 mg, 2.22 mmol) were dissolved in 15 mL of DMF, and the solution was cooled to 0 °C. To a prepared solution, compound **L2** (1.00 g, 1.48 mmol) and EDC × HCl (426 mg, 2.22 mmol), and after 5 min, TEA (1.0 mL, 7.39 mmol), were added. The reaction mixture was stirred at RT for 24 h. Compound **1** was precipitated by concentrating the reaction mixture in vacuo followed by cooling to 0 °C and adding cold water (15 mL) dropwise. The precipitate was filtered off under low pressure, washed with water, and dried in air. A total of 1.40 g (90%) of compound **1** of high purity was obtained and used without further purification.

**^1^H NMR (400 MHz, DMSO-*d*_6_) *δ*/ppm:** 8.51 (d, *J* = 7.46 Hz, 1H); 8.17 (d, *J* = 8.19 Hz, 1H); 8.08 (d, *J* = 7.92 Hz, 1H); 8.07 (d, *J* = 8.04 Hz, 1H); 8.00 (d, *J* = 8.27 Hz, 1H); 7.41 (d, *J* = 8.84 Hz, 1H); 7.35–6.92 (m, 35H); 4.91 (d, *J* = 3.48 Hz, 2H); 4.66–4.44 (m, 5H); 4.24–4.14 (m, 1H); 3.57 (s, 3H); 3.09–2.88 (m, 6H); 2.87–2.65 (m, 5H); 2.60 (dd, *J* = 13.78 Hz, 11.06 Hz, 1H).

**^13^C NMR (400 MHz, DMSO-*d*_6_) *δ*/ppm:** 172.10, 171.63, 171.43, 171.06, 156.11, 138.55, 138.02, 137.99, 137.92, 137.45, 129.73, 129.66, 129.61, 129.50, 128.75, 128.51, 128.44, 128.40, 128.11, 127.83, 127.04, 126.73, 126.62, 65.62, 56.50, 54.08, 54.06, 54.01, 53.95, 52.30, 38.13, 38.05, 37.92, 37.14.

**HRMS (ESI^+^) *m*/z** C_63_H_64_N_6_O_9_Na [M + Na]^+^ calcd: 1071.4632, found: 1071.4614.

### 3.3. Synthesis of l-phenylalanyl-l-phenylalanyl-l-phenylalanyl-l-phenylalanyl-l-phenylalanyl-l-phenylalanine Methyl Ester (L3)

Compound **1** (1.40 g, 1.33 mmol) and NH_4_HCO_2_ (335 mg, 5.32 mmol) were added to a suspension of 5% Pd/C (140 mg) and MeOH (100 mL). The reaction mixture was stirred under reflux for 2 h then cooled to RT, filtered over a celite pad, and dried in vacuo. The crude solid was purified on a column chromatography with silica gel and TEA:MeOH:DCM (0.3:5:94.7) as an eluent to obtain 930 mg (76%) of pure compound **L3**.

**^1^H NMR (400 MHz, DMSO-*d*_6_) *δ*/ppm:** 8.51 (d, *J* = 7.45 Hz, 1H); 8.16 (d, *J* = 8.24 Hz, 2H); 8.08 (d, *J* = 8.10 Hz, 1H); 7.93 (d, *J* = 6.40 Hz, 1H); 7.30–7.00 (m, 30H); 4.64–4.43 (m, 5H); 3.57 (s, 3H); 3.30 (dd, *J* = 8.89 Hz, 4.39 Hz, 1H); 3.10–2.65 (m, 11H); 2.41 (dd, *J* = 13.52 Hz, 8.76 Hz, 1H).

**^13^C NMR (400 MHz, DMSO-*d*_6_) *δ*/ppm:** 174.07, 172.09, 171.42, 171.22, 171.06, 171.03, 139.01, 138.17, 138.00, 137.92, 137.81, 137.44, 129.83, 129.75, 129.65, 129.50, 128.75, 128.60, 128.50, 128.45, 128.33, 127.04, 126.73, 126.64, 126.60, 56.44, 54.14, 54.08, 53.92, 53.32, 52.30, 40.90, 38.29, 38.13, 38.07, 37.98, 37.14.

**HRMS (ESI^+^) *m*/z** C_55_H_59_N_6_O_7_Na [M + H]^+^ calcd: 915.4445, found: 915.4422.

### 3.4. Physicochemical Measurements

**Materials.** The salts used for the determination of peptide affinity towards ionic species were tetraethylammonium chloride (TEACl, Sigma Aldrich, St. Louis, MO, USA, 98%); tetrabutylammonium bromide (TBABr, Sigma Aldrich 99.0%); tetrabutylammonium hydrogen sulfate (TBAHSO_4_, Sigma Aldrich, 99.0%); tetrabutylammonium dihydrogen phosphate (TBAH_2_PO_4_, Sigma Aldrich, 99.0%); and tetrabutylammonium nitrate (TBANO_3_, Sigma Aldrich, 97%). The solvents used in the physicochemical measurements (MeCN (J. T. Baker, HPLC Gradient Grade); CD_3_CN (Eurisotop, Cambridge, UK, 99.80% D); DMF (Supelco brand of Sigma Aldrich, for spectroscopy); and DMF-*d*_7_ (Eurisotop, 99.50% D)) were used without further purification.

**Spectrofluorimetry.** Spectrofluorimetric titration experiments were carried out at (25.0 ± 0.1) °C by means of an Agilent Cary Eclipse spectrofluorimeter (Santa Clara, CA, USA) equipped with a thermostatting device. The spectral changes of linear peptide solutions (*V*_0_ = 2 cm^3^, *c*_0_ = 5 × 10^−5^ to 2 × 10^−4^ mol dm^−3^) were recorded upon the stepwise addition of a salt solution (*c*_0_ = 1 × 10^−3^ mol dm^−3^ to 0.2 mol dm^−3^) directly into the measuring quartz cell (Agilent, QS, *l* = 1 cm). Spectra were sampled at 1 nm intervals, with an integration time of 0.4 s. The titrations for each ligand/ion system were repeated in triplicate. The obtained spectral data were processed by the HypSpec program (Protonic Software, Leeds, UK) [33,34].

**^1^H NMR Spectroscopy.**^1^H NMR titration spectra were recorded by means of a Bruker Ascend 400 spectrometer with TMS as an internal standard. The spectral changes of ligand solutions (*V*_0_ = 0.5 cm^3^, *c*_0_ = 1 × 10^−4^ to 1 × 10^−3^ mol dm^−3^) were recorded upon the stepwise addition of a salt–ligand solution mixture (*c*_0_ = 8 × 10^−4^ to 0.5 mol dm^−3^) directly into the NMR tube containing the peptide solution. The obtained spectral data were processed by the HypNMR program (Protonic Software, Leeds, UK) [35,36].

**Circular Dichroism.** CD titration spectra were recorded on a JASCO J815 spectrophotometer (Halifax, NS, Canada) at room temperature in the 200–300 nm wavelength range, with a 0.2 nm data pitch and scanning speed of 200 nm/min; 2 scans were accumulated for each spectrum. In all titrations, a quartz cell with a 1 cm optical path length was used, and the initial volume of the titrant solution was 2 mL. Spectra were collected after the stepwise addition of an anion salt solution (*c*_0_ = 3 × 10^−3^ to 0.3 mol dm^−3^) to the reaction mixture containing the peptide (*c*_0_ = 1 × 10^−4^ to 5 × 10^−4^ mol dm^−3^). The obtained data were processed by the HypSpec program [33,34].

**Microcalorimetry.** Microcalorimetric measurements were conducted by means of an isothermal titration calorimeter, Microcal VP-ITC (Malvern, Worcestershire, UK), at 25.0 °C. The enthalpy changes were recorded upon stepwise additions of a salt solution (*c*_0_ = 7 × 10^−3^ to 0.05 mol dm^−3^) into the solution of the peptide ligand (*c*_0_ = 1 × 10^−4^ to 1 × 10^−3^ mol dm^−3^). The heats measured in the titration experiments were corrected for the heats of the titrant dilution obtained by blank experiments. Thermograms were processed using the Microcal OriginPro 7.0 program (OriginLab Corporation, Northampton, MA, USA). The dependence of successive enthalpy changes on the titrant volume was processed by a non-linear least-squares fitting procedure, and in the case of two or more equilibria, the HypDH program was used [37]. Measurements were performed in triplicates. In the determination of the thermodynamic parameters of chloride complexation in DMF, a peptide solution was used as a titrant and a salt solution as a titrand. In this case, the experiment was carried out once due to the high consumption of peptides. ∆(∆*H*_m_) denotes a molar successive enthalpy change after a single titrant addition and equals to a successive enthalpy change ∆(∆*H*) divided by the amount of titrant.

**Molecular dynamics simulations.** The molecular dynamic simulations were carried out by means of the GROMACS package (version 2020.5, University of Groningen, Royal Institute of Technology, Uppsala University) [38,39,40,41,42,43]. Intramolecular and nonbonded intermolecular interactions were modelled by the OPLS–AA (Optimized Potential for Liquid Simulations—All Atom) force field [44]. The initial structures of the peptide complexes were prepared by the placement of an anion in the proximity of amide hydrogens, followed by the *NVT* vacuum MD simulations. The free ligands and their anion complexes were solvated in a cubical box with a side length of 6.5 nm with periodic boundary conditions, containing approximately 2900 MeCN or 1900 DMF molecules. The solvent boxes were equilibrated prior to the solvation of the peptide ligands and the corresponding complexes. The solute concentration in such a box was about 0.01 mol dm^−3^. We were unable to obtain stable structures of the anion dimer complexes due to limitations of the force field parameters or the classical level of theory used for the intermolecular interactions. During the simulations of the systems, the TBA^+^ ion was included to neutralize the box. This counterion was held fixed at the box periphery, whereas the complex was initially positioned at the box center. We monitored the distance between the restrained TBA^+^ cation and complexed anions during all simulations, and we observed no ion association at any point. In all simulations, an energy minimization procedure was performed, followed by 50.5 ns of an *NpT* production simulation. All simulations were performed without repeating. The first 0.5 ns of the production simulation were discarded in the data analysis. The integrator used for the propagation, and also for the temperature, control was a stochastic dynamic algorithm with a time step of 1 fs [45]. The temperature was kept at 298 K during simulation. The pressure was kept at around an average value of 1 bar with the Parrinello–Rahman barostat [46,47]. The cutoff radius for nonbonded van der Waals and short-range Coulomb interactions was 15 Å. Long-range Coulomb interactions were treated by the Ewald method as implemented in the PME (Particle Mesh Ewald) procedure [48]. The representative molecular structures of the peptide–ion complexes were obtained by PCA (Principle Component Analysis) on a coordination matrix, whose rows contained distances between the anions (or oxygen atoms of anions) and hydrogen atoms of the amide or amine moiety during simulation. Angles between anions (or oxygen atoms of anions), hydrogen atoms of the amide or amine groups, and their respective nitrogen atoms were added to the coordination matrix as well. The chosen structures were closest to the centroids of the most populous clusters in space defined by the first two principal components. The coordination matrix of free peptides was constructed of distances between hydrogen atoms of the amide or amine groups and carbonyl oxygen atoms. The angles between the hydrogen atoms of the amide or amine groups, their respective nitrogen atoms, and carbonyl oxygen atoms were also used. Figures of molecular structures were created using the VMD software (version 1.9.2, University of Illinois) [49].

## 4. Conclusions

In this work, we demonstrated that methyl esters of tetra-(**L1**), penta-(**L2**), and hexaphenylalanine (**L3**) peptides can act as versatile anion receptors in organic solvents. The formation of peptide–anion complexes in MeCN and DMF was thermodynamically characterized by the determination of stability constants and also in terms of enthalpic and entropic contributions to the complexation process. The obtained thermodynamic parameters were correlated to the peptide chain length with respect to the energetics of binding and entropic contributions of chain reorganizations. All three peptides bind chloride, bromide, hydrogen sulfate, dihydrogenphosphate, and nitrate anions by forming complexes of 1:1 stoichiometry. In the cases of hydrogen sulfate and DHP, complexes of higher stoichiometries were observed as well (1:2 and 2:1 sandwich complexes). The complexation is exothermic and enthalpy driven in most cases, although in some instances, an entropic contribution is equally important or even dominant. The structural characteristics of peptide–anion complexes were assessed by molecular dynamic simulations. The anions are bound by nearly all peptide backbone amide protons and one *N*-terminal amine proton by hydrogen bonding, and in the course of the reaction, the conformation of the peptide backbone changes from elongated to quasi-cyclic in all 1:1 complexes. The anion binding and conformational changes studied in this work are crucial for the synthesis of cyclopeptides where head-to-tail macrocyclizations of linear analogs occur. Also, the anion–receptor properties of oligopeptides open the possibility of their application as ionophores in transmembrane anion transport.

## Data Availability

Data is contained within the article.

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
