# Peer review of "Anion-Binding Properties of Short Linear Homopeptides"

_ijms, 2024, doi:10.3390/ijms25105235_

Round 1

Reviewer 1 Report

Comments and Suggestions for Authors

The manuscript titled “Anion Binding Properties of Short Linear Homopeptides” was prepared to carried out a comprehensive thermodynamic and structural study of the complexation affinities of tetra-, penta- and hexaphenylalanine linear peptides towards several inorganic anions in acetonitrile (MeCN) and N,N-dimethylformamide (DMF). It was able to provide a insight into the direct interaction between peptide and anion. The data of the manuscript basically supports the conclusion, but there are still some problems that need to be explained before the official publication.

1)For the part of introduction, it is necessary to demonstrate the value of explore the interaction between peptide and anion.

2)For the part of molecular dynamic simulation method, the author should introduce the ratio and quantity of peptides and anions as well as the standard of this setting. In addition, the total simulation time and repeat times should also provided.

3)At present, the systems were neutralized with TBA+ion during the simulation, is there any influence of TBA+ion on the interaction between peptide and anion was observed ?

4)The author detected the interaction between peptide with different anion based on molecular dynamic simulation, is it necessary to compare the preference of peptide for anion ? In addition, the author should provide the detail of how to obtain representative conformations for analysis.

5)Besides the hydrogen bond and conformational analysis, other simulation analyzes are also necessary.

Reviewer 2 Report

Comments and Suggestions for Authors

This paper by Modrusan describes a systematic binding study characterizing the interaction of three linear all-phenylalanine peptides of different lengths with a selection of anions in acetonitrile and DMF. The authors report the binding constants of the complexes of the three peptides with halides and a selection of oxoanions determined by different methods (fluorescence, circular dichroism, NMR, calorimetry). The binding constants are in good agreement for each complex studied, indicating that 1:1 complexes are usually formed. Molecular dynamics shows that the peptides usually wrap around the anions, bringing the end groups in close proximity, suggesting that anions could be suitable templates to mediate macrocyclization reactions to prepare cyclopeptides.

The work is interesting and provides a set of useful binding constants to predict the interactions of anions with peptides in nonaqueous solutions. The manuscript therefore deserves to be published, provided that the following revisions are taken into account:

The authors used the peptides with the N-terminus unprotected and the C-terminus protected. Please explain this decision.

Reviews on anion binding to peptide-derived receptors could be cited, such as https://doi.org/10.1039/b810531f and similar recent ones.

Given that the peptides were used as free amines, it is striking that only acidic anions (hydrogen sulfate and dihydrogen phosphate) form 2:1 complexes with the peptides in acetonitrile (and dihydrogen phosphate also in DMF). Is it possible that the first anion mediates the protonation of the N-terminal amino group in the peptides? Please provide information in this regard.

Only the 1:1 complexes were calculated, but some anions form higher complexes. Could structural information be provided for these complexes as well (the authors speculate on the binding of anion dimers at one point)?

Comments on the Quality of English Language

Correct typo in line 297 (exergonic).

Round 2

Reviewer 1 Report

Comments and Suggestions for Authors

The authors have address all my concerns and the manuscript is suitable for publication.